# Zincon-Modified CNTs Electrochemical Tool for Salivary and Urinary Zinc Detection

**DOI:** 10.3390/nano12244431

**Published:** 2022-12-13

**Authors:** Daniela Vieira, Jérôme Allard, Kathleen Taylor, Edward J. Harvey, Geraldine Merle

**Affiliations:** 1Department of Experimental Surgery, Faculty of Medicine, McGill University, 1650 Cedar Avenue, A7-117, Montreal, QC H3G 1A4, Canada; 2Department of Chemical Engineering, Polytechnique Montreal, J.-A.-Bombardier Building, Office 2067, Montreal, QC H3C 3A7, Canada; 3Department of Chemical and Biomolecular Engineering, Georgia Institute of Technology, 311 Ferst Dr, Atlanta, GA 30318, USA

**Keywords:** electrochemical detection, carbon nanotube, zinc quantification, zincon, complexation, saliva, urine

## Abstract

Recently, the abnormal level of zinc emerged as a powerful indicator or risk factor for metabolic, endocrine, neurodegenerative and cardiovascular diseases, including cancer. Electrochemical detection has been explored to quantify zinc in a precise, rapid, and non-expensive way; however, most of the current electrochemical systems lack in specificity. In this work we studied a highly selective and sensitive electrochemical method to detect quickly and reliably free zinc ions (Zn^2+^). The surface of the working electrode was modified with zincon electropolymerized on carbon nanotube (CNT) to enable the binding of zinc in complex body fluids. After being physicochemically characterized, the performances of the zincon-CNT complex was electrochemically assessed. Square Wave Voltammetry (SWV) was used to determine the calibration curve and the linear range of zinc quantification in artificial saliva and urine. This zincon- CNT system could specifically quantify mobile Zn^2+^ in salivary and urinary matrices with a sensitivity of ~100 ng·mL^−1^ and a limit of detection (LOD) of ~20 ng·mL^−1^. Zincon-modified CNT presented as a desirable candidate for the detection and quantification of free zinc in easily body fluids that potentially can become a diagnostic non-invasive testing platform.

## 1. Introduction

Metallomics is a systematic study on the identification and concentration of essential metals for life processes. It is becoming fundamental for the understanding of biological systems. Metals are crucial for bodily function and are directly involved on a variety of biochemical processes such as osmotic regulation, catalysis, metabolism, and cell signaling [1,2,3]. Despite their vital role, metal deficiency or overexpression are associated with several health conditions. The monitoring of the concentration of some essential metals may help to identify, track progression, and evaluate the impact of therapeutic agents in a variety of diseases [2,3,4,5].

Zinc (Zn) is one of the most abundant trace elements in the body and it is essential for more than 300 enzyme functionalities [6,7,8]. Zinc plays a crucial role to intracellular communication, immune system, homeostasis, apoptosis, DNA repair and replication, balancing oxidative stress, and the aging process [7,9,10]. The deficiency or excess of zinc is potentially a powerful indicator of risk factor for various health issues, including metabolic, endocrine, neurodegenerative, cardiovascular diseases, and cancer [4,5,11]. Salivary and/or urinary zinc level is abnormal in a variety of cancer [8,12], including prostate [13,14,15], oral [16,17], breast [15,18,19,20,21], pancreatic [15,22], brain [23], lung [24] and bladder [25] (Table 1).

The detection and quantification of zinc through saliva and/or urine is ideal because it enables non-invasive sampling, high volume collection, and ease repeatability of measurements when compared to blood, plasma, or tissues specimens [15,26]. Common applied methods to quantify zinc involve atomic absorption spectroscopy, inductively coupled plasma mass spectrometry, and Raman spectroscopy [4,27]; however most tests utilize sizable machinery and highly qualified personnel, are time consuming, and costly. Salivary and urinary zinc could be easily accessible through electrochemical techniques with a user-friendly, precise, and specific quantification without invasive or costly testing.

The most common electrochemical systems applied to detect and quantify zinc involve mercury and bismuth electrodes [27,28,29,30,31]. Besides their great ability to detect zinc, mercury is extremely toxic, and bismuth is usually applied as a co-deposited film, being invalid for many applications [28,29,30]. Other materials such as carbon-based electrodes (carbon nanoparticles, carbon nanotube e, graphene), show very low limit of detection (LOD), reaching nM [27,28,32]; however, carbon alone does not present specificity and barely have been tested in body fluids for zinc detection and quantification. One of the emergent strategies to assure the specificity of carbon-based electrodes towards zinc detection is through the functionalization of carbon with molecules that allow complexation with zinc [33,34]. EDTA [35], amino acids [36] and zincon [37] are promising mediators for electron transfer by complexing and de-complexing zinc [33,34]. In this way, zincon stands out as metal chelator (Pb, Cu and Zn) and demonstrates great ability on the detection of metallic content in environmental and biological samples by colorimetric methods [37,38,39]. Recently, zincon has been explored as a complex agent in electrochemical systems for heavy metals quantification (Zn and Pb) [37,39]. Most of studies applied zincon electro-polymerization to modify the carbon-based electrodes [40,41,42,43], which must be extremely well controlled to avoid the blockage of the active sites [44,45]. A zincon exfoliated graphite electrode, to quantify free zinc ions in prostate fluid, was built via the π- π stacking interaction [37]. Unfortunately, this novel system presented a high LOD and poor stability probably because of the instability of the non-covalent bind between the ink and the exfoliated graphite. New alternatives must be investigated to ensure stability and conductivity of zincon carbon-based electrodes.

In this work, we engineered a fast, non-invasive, precise, and specific electrochemical zinc sensor based on the covalent grafting of zincon on multi-walled carbon nanotubes (CNTs). CNTs offer excellent intrinsic properties such as high surface area, chemical stability and high electrical conductivity (10^6^–10^7^ S/m) [46], becoming one of the most attractive nanomaterials in electrochemical sensing [47,48,49]. Through a better immobilization and stabilization of zincon on CNTs, we hypothesized that the system’s performance will be strongly enhanced and that the electron transfer will be facilitated leading to an increased LOD and specificity and a minimization of interferences—a critical feature given bodily fluids have thousands of metabolites [37,50]. Among the different immobilization methods [49], the covalent grafting was chosen via an in situ radical polymerization using potassium persulfate as a radical initiator, followed by refluxing with concentrated hydrochloric acid [47]. The resulting zincon-modified CNT sensor was physiochemically characterized and tested in buffer and artificial biofluids, i.e., saliva and urine, demonstrating great ability to quantify zinc when simulating its concentration for different types of cancer.

## 2. Materials and Methods

### 2.1. Materials and Chemicals

Zincon monosodium salt and glassy carbon electrodes (GCE) were purchased from Alfa Aesar, Haverhill, MA, USA. Copper sulfate (99.99%), magnesium sulphate (99.99%), manganese chloride (97%), iron sulfate heptahydrate (98%), calcium chloride (93%), anhydrous, cobalt chloride hexahydrate (98%), and nickel sulfate hexahydrate (99%) were purchased from Sigma Aldrich. Carbon nanotubes (CNTs) were purchased from Sigma-Aldrich, Darmstadt, Germany. Artificial saliva (#1700-0305) was purchased from Pickering laboratories, Park Way, Mountain View, CA, USA. Artificial urine (control #IS5080) was purchased from Aldon Corporation, Avon, NY, USA (Table 2). All the other chemicals and materials used in this work were purchased from Fisher Scientific, Scotia Court, ON, Canada.

### 2.2. Zincon-Modified CNT

Zincon-modified CNT was prepared adapting the protocol of Zhang et al. [47]. Briefly, 1 g of zincon monosodium salt (100%, Alfa Aesar, Haverhill, MA, USA) was dissolved in 100 mL of ultrapure water (Milli-Q, Merck, Darmstadt, Germany), and then 50 mg of multi walled carbon nanotubes (CNTs) (>98%, Sigma-Aldrich, Oakville, ON, Canada) was added. After sonication for 1 h, the black solution was stirred for another 3 h at room temperature. Afterwards, the mixture was purged with nitrogen gas for 30 min to remove oxygen and 0.25 g of potassium persulfate (>99%, Fisher Scientific, Scotia Court, ON, Canada) was added into as a radical initiator. The solution was stirred continuously at 70 °C for 3 h, followed by at 85 °C for 1 h. The solution was filtered through a 0.22 µm nylon membrane (GVS, Fisher Scientific, Scotia Court, ON, Canada), washed several times with ultrapure water and redissolved in concentrated hydrochloric acid (HCl) (35–39%, Fisher Scientific, Scotia Court, ON, Canada) for 12 h at 100 °C. Finally, after being filtered through a 0.22 µm nylon membrane, the zincon-modified CNT product was washed several times with ultrapure water until pH 7 is reached. The black powder was dried overnight at 45 °C in incubator (Blue M 100A, Blue Island, IL, USA). Solution containing 1 mg·mL^−1^ zincon-modified CNT in ethanol (89–91%, Fisher Scientific, Scotia Court, ON, Canada) were prepared and stored. Solutions containing 1 mg·mL^−1^ of pristine CNT was also prepared. Glassy carbon electrodes (GCE) (5 mm, Alfa Aesar, Haverhill, MA, USA) were polished with 0.05 µm Al_2_O_3_ (Fisher Scientific, Scotia Court, ON, Canada) suspension to achieve a shiny surface. GCEs were cleaned by sonication in 10% H_2_SO_4_ (98%, Fisher Scientific, Scotia Court, ON, Canada), 50% acetone (≥99.5%, Fisher Scientific, Scotia Court, ON, Canada) and ultrapure water each for 10 min, successively. The electrodes were dried at room temperature. Finally, 10 µL of the zincon-modified CNT solution was dropped onto the cleaned GCE and dried at room temperature. Pristine CNT was also performed following the same procedure.

### 2.3. Materials Characterization

The morphology of CNT and zincon-modified CNT electrodes was investigated by Scanning Electron Microscopy (SEM, Inspect F50, FEI Company, Hillsboro, OR, USA). ImageJ software (National Institutes of Health, Stapleton, New York, United States) was applied to measure the diameter of CNT and zincon-modified CNT (*n* = 10). Fourier transform infrared spectroscopy (FT-IR—PerkinElmer) was carried out in the wavenumber range of 4000 to 400 cm^−1^ to confirm the grafting of zincon on CNT. Electrochemical experiments were performed using a potentiostat (VersaSTAT 4, Princeton Applied Research, Oak Ridge, TN, USA) with a three-electrode system cell, where GCE, CNT, zincon-modified CNT were used as work electrodes, Pt wire as counter electrode and saturated calomel electrode (SCE) as reference electrode. The electrochemical behavior was tested in different solutions, such as, 1 mM K_4_Fe(CN)_6_ in a 0.1 M KCl (99%, Fisher Scientific, Scotia Court, ON, Canada), buffer (0.1 M Tris-HCl pH 7.5), artificial saliva (Pickering laboratories, Park Way, Mountain View, CA, USA) and artificial urine (Aldon Corporation, Avon, NY, USA) solutions in presence and/or absence of zinc (Zinc acetate, anhydrous, >99.9%, Fisher Scientific, Scotia Court, ON, Canada). Note that saliva and urine solutions were diluted by 50% with buffer (pH 7.5). Cyclic voltammetry (CV) was carried out in the presence and absence of zinc. The potential was scanned from −0.8 V to −1.3 V (vs. SCE) at scan rate of 50 mV·s^−1^. Square wave stripping voltammetry (SWV) was performed to characterize the electrochemical detection of Zn using CNT and zincon-modified CNT electrodes. The potential of deposition was selected at −1.40 V (vs. SCE), applied for 300 s, and followed by a SWV, from −1.4 V to −0.7 V (vs. SCE), at 10 Hz, amplitude of 25 mV, and step potential of 40 mV. A pre time (adsorption time) of 180 s was applied (with no potential applied, electrical connections were disconnected to the electrode). All analysis were performed in triplicates (*n* = 3). Linear fit was obtained using OriginPro (OriginLab Corporation, version 2018G, Northampton, MA, USA) software. Different parameters of zinc deposition (time and potential) were studied to verify the best performance. Zincon-modified CNT electrodes were tested using different concentrations of zinc solutions (from 0 to 1000 ng·mL^−1^). Experiments were performed in triplicates.

Limit of detection (LOD) was calculated according to the Formula (1):LOD = 3 × (SD/S)(1)
where, SD is the standard error intercept and the S is the slope of the calibration curve (S), both extracted from origin software after linear regression.

### 2.4. Statistics

The data were analyzed using OriginPro (OriginLab Corporation, version 2018G) and presented as mean ± SD. Repeatability standard deviation was obtained by the square foot of the variance. Stability was calculated according to Formula (2):(2)$$Stability=100\%-\left|Relative\;deviation\right|$$
where $relative\;deviation=\left(\frac{different\;time\;detection\;value-baseline}{baseline}\right)\times 100$.

## 3. Results and Discussion

### 3.1. Materials Characterizations

In this work, we proposed a covalent polymerization of zincon on CNTs to maximize the specificity and ability of the modified CNTs for the quantification of free zinc. As noted above the synthesis was performed in water solution, adding potassium persulfate (KPS) as a free radical initiator, followed by HCl reflux. Zincon not only acted as a reaction monomer but also as a dispersant of pristine CNT in water because of the presence of hydrophilic and hydrophobic chains, dismissing the use of additional dispersants [47,53]. CNTs tend to agglomerate due to the van the der Walls forces. Here, the presence of sulfonated group in zincon assures hydrophilicity and electrostatic repulsion to overcome the van der Waals attraction of CNT, while the hydrophobic part form π- π stacking interaction with the CNT [47]. The covalent functionalization enables strong interfacial interactions on CNTs. In addition, it is expected a better dispersion, facilitating the flow of electrons and enhancing the interaction between the analyte and the electrode surface [54,55]. The suggested mechanism of the covalent grafting facilitated by KPS and, the morphology of CNT and zincon-modified CNT is demonstrated in Figure 1A. The surface morphologies of zincon-modified CNT, and CNT electrodes are shown in Figure 1B,C. Low magnification SEM analysis showed a higher quality dispersion of CNT onto electrodes as less aggregates are observed on the zincon-modified CNT electrode surface. As mentioned, this is probably the result of the sulfonic groups on benzene ring that ensure adequate electrostatic repulsions [47,53]. In Figure 1B, the coating appears to be denser for CNT- compared to the zincon-modified CNTs coating. The pristine CNTs are tangled together, due to the van der Waals attractions between tubes. After grafted with zincon, the materials because of the sulfonic group of the zincon, have a much better dispersibility than pristine CNTs. At higher magnification, the morphology of the CNTs appears to be unchanged, confirming that after the zincon polymerization treatment, no obvious alterations of the tubular structure of CNTs could be observed (Figure 1B). The CNT fiber’s diameter was measured using ImageJ software (*n* = 10). Untreated CNTs and zincon-modified CNTs presented similar diameters of 18.07 ± 3.15 nm and 18.54 ± 1.42 nm, respectively, indicating that the graft of zincon does not affect the tubular structure of CNTs. Similar effect has been already observed using same chemical treatment to graft poly(p-styrenesulfonic acid) onto multi-walled carbon nanotubes [47].

The successful grafting of zincon on CNT was confirmed by FT-IR (Figure 2A) and cyclic voltammetry (CV) (Figure 2B). The FT-IR spectrum of zincon-modified CNT (Figure 2A) showed, as expected, additional peaks compared to pristine CNTs at ~1660 cm^−1^ and at ~1530 cm^−1^ are attributed to C=N stretching and N=N stretching, respectively. The peaks at ~1500 cm^−1^ and ~1350 cm^−1^ are due to the benzene ring vibration of the zincon [40,56,57]. The broad band in the region at ~1250 cm^−1^ to 750 cm^−1^ could be attributed to the stretching of C–C and C–O in the carbon after the treatment with concentrated hydrochloric acid for 12 h [56]. It should be noticed that the as-received CNTs were purified by the manufacturer and partial oxidation during purification by the manufacturer can result in the presence of groups on the surface of pristine carbon nanotube. The peak ~3700 cm^−1^, for both CNT and zincon-modified CNT, could be related to the absorption of water molecules as a result of an O-H stretching mode of hydroxyl groups [47,56]. The peaks at ~2670 cm^−1^, 2320 cm^−1^, 2097 cm^−1^, and at 1990 cm^−1^ could be attributed to the C=C asymmetric stretching in graphite-like CNT structure [58].

The cyclic voltammogram of zincon-modified CNT (Figure 2B) clearly showed the reduction (~0 V and −0.6 V vs. SCE) and oxidation (−0.4 V and ~0.1 V vs. SCE) peaks for zincon [43] and demonstrated a good stability after 10 cycles of CV (Figure 2C).

Carbon structures have been previous modified with zincon through the electropolymerisation of poly-zincon films [40,41,42,43] or π-π stacking interaction [37]. However, as mentioned before, both techniques may suffer from loss of conductivity because of the enclosed surface and/or the zincon attachment instability- signal loss after few measurements if the polymer attachment is not well controlled [44,45]. The covalent polymerization grafting process assures a more robust and stable connection between the zincon compound and the CNTs in comparison to noncovalent procedures, generating a more effective electrode interface [59,60,61,62].

### 3.2. Electrochemical Behavior

Cyclic voltammetry of the ferrocyanide/ferricyanide redox couple (Fe(CN)_6_^3−/4−^) was used to verify the grafting of zincon on CNT (Figure 3A). As expected, an increase of ~42% in peak current density (I_pa_ = ~10 µA), and a decrease of ~53% in peak width (ΔE_p_ = ~77 mV) can be observed in the Fe(CN)_6_^3−/4−^ voltammograms (Figure 3A), after deposition of CNTs on GCE because of the electron transport kinetics at the electrode. Successive attachment of zincon onto CNTs leads to a drop of ~58% in peak current density (I_pa_ = ~7 µA) and growing of ~47% in peak width (ΔE_p_ = ~145 mV), attributed to a loss of conductivity. These results were consistent with FTIR and previous CV, confirming the successful grafting of zincon on CNT.

Besides the reduction of the redox currents when compared to CNTs, in Figure 3B it is clear that zincon-modified CNT influences the quantification of zinc. CVs response in buffer for GCE, CNT and zincon-modified CNT showed an increase on both reduction and oxidation peaks when zincon was applied. A shift in the reduction potential was observed for the different electrodes (Figure 3B). The unmodified GCE and pristine CNT showed reduction peaks at ~−1.3 V and ~−1.25 V, respectively; whereas the zincon-modified CNT showed reduction peak at ~−1.34 V. Pristine CNT electrode showed the lowest peak separation (~0.23 V), indicating higher reversibility of Zn^0/^Zn^+2^ and higher number of active sites [63]. However, the peak current intensity is lower than zincon-modified CNT. The reduction current intensity increased by ~140 µA for the modified electrode when compared to pristine CNT, suggesting the greater capability of the compound for the quantification of zinc.

The mechanism behind the detection is probably attributed to the zincon quadridentate ligand property, coordinating zinc ions with two oxygens (OH and COOH) and two nitrogens (N=N and NH) [37,50,64]. Further, the complexed Zn^+2^ is preconcentrated onto the electrode surface at −1.4 V (vs. SCE) for 300 s, allowing the reduction for Zn^0^. Followed by a potential scan (from −1.3 V to −0.7 V), the zinc stripped back to the solution (Zn^0^ → Zn^+2^) and the current intensity is measured in function of concentration of zinc (Figure 4). Note that the pH is a very important parameter to assure the complexation. Zincon can also complex with other metals, such copper, iron, cobalt, and nickel; however, the optimized pH for these metals and zincon complexation is at pH~5 [64,65]. The optimized zincon-zinc complexes occur in pH > 7.0, being inefficient for pH < 6.0 [37,64]. To ensure the complexation between zincon-zinc, a pre-time of 180 s was used before metal deposition and the stripping steps.

We first studied the ability of zincon-modified CNT on the quantification of free zinc in a buffer setup. The linear range and limit of detection (LOD) of zinc by zincon-modified CNT was determined by square wave voltammetry (SWV), varying the concentration of zinc from 0 to 1000 ng·mL^−1^, in a Tris HCl (pH 7.5) buffer solution. Figure 5 shows the increase of the ability for the quantification of zinc when zincon was grafted on CNT.

Figure 5A shows the CV of zincon-modified CNT in presence and absence of zinc. The zinc reduction and oxidation peaks, at ~−1.3 V and ~−1 V (vs. SCE), respectively, were found in a well-defined shape. As expected, no peaks were found in buffer without addition of zinc. In order to confirm the improvement of zinc detection by the complexation mechanism with zincon, we compared the current responses for CNT and zincon-modified CNT in presence of 1 µg·mL^−1^ of zinc (Figure 5B). The zincon-modified CNT electrode showed more efficacy for the detection of zinc, where the current intensity enhanced by ~40 µA when compared to pristine CNT (160 µA and 120 µA, respectively). Further, LOD was determined by SWV for zincon-modified CNT in a buffer solution varying the concentration of zinc. Figure 5C shows the linear response of current intensity in function of the concentration of zinc. Zincon-modified CNT system was able to quantify 50 ng·mL^−1^ of zinc and presented a LOD of 15.4 ng·mL^−1^. A zincon-modified exfoliated graphite electrode based on π stacking was designed in order to quantify free zinc in prostate fluids [50]. The system could only quantify 250 ng·mL^−1^ of free zinc whereas our electrode could quantify 50 ng·mL^−1^, 5-fold less than the π stacking system, probably attributed to the stronger connection between zincon and carbon structure via a covalent in situ polymerization. It has been previously shown using UV–vis spectroscopy that zincon in solution with same quantity of bivalent metals such as Zn, Cu, Cd, Pb, Co, Ca, Ni, Ba, Mg and Mn, only changes in color with Cu^2+^ similarly to Zn^2+^. Based on this observation [37], an interference study was performed on the zincon-modified CNT electrode (Figure 6). We chose Ca, Cu, Co, Mg, Mn, and Fe since they are the most common metals in body and quite often presented in body fluid samples [5,66] and may interfere with Zn^+2^ detection. The detection of these bivalent metals was performed using CV and SWV, and as it can be seen, peak of Zn^+2^ (about potential −1 V) is well separated from other metals (Figure 6).

SWVs showed that the presence of other metals did not affect the peak heights, confirming that all the tested metallic ions exhibited no apparent interference in Zn^+2^ detection. To further study, interferences and contaminants, the viability of the zincon system reported here as a sensor was verified in 2 different artificial body fluids—urine, and saliva. Urine and saliva have been specifically explored as diagnostic fluids, because they are a source of biomarkers, allowing non-invasive analysis [15,26]. Figure 7 shows the CV and SWV for zincon-modified CNT in artificial urine and saliva solutions.

The ability of zinc quantification in body fluids was studied by CV and SWV. In artificial urine (Figure 7A) a background reduction peak (at ~−1.4 V) was observed when zinc was absent, this is probably because of the elements present in the artificial urine (Table 2). Stronger and well-defined reduction and oxidation peaks were observed in presence of 0.5 mg·mL^−1^ of zinc. Same behavior could be seen by SWV, where in absence of added zinc a background current was observed, increasing the intensity as the concentration of zinc increased. Similar behavior was observed in artificial saliva with the increase of concentration of zinc; however, no background signals were found in absence of zinc (Figure 7B). The zincon-modified CNT showed affinity for the detection and quantification of free zinc when simulating both body fluids, presenting linear detection from 125 ng·mL^−1^ and LOD of ~30 ng·mL^−1^ and ~20 ng·mL^−1^ for urine and saliva, respectively. We have tested repeatability of the system by the repeatability standard deviation, which can be measured by the square root of the variance. For the buffer, saliva and urine, the repeatability standard deviation was ±7.07 µA, ±0.55 µA and ±3.28 µA, respectively. The system showed stability of ~65% after 1 year. The designed zincon-modified CNT system not only showed considerable sensitivity but also presented the advantages of simplicity and lower LOD.

Salivary and/or urinary zinc levels are a powerful indicator of risk factor for a variety of cancer as shown in Table 1. The designed zincon-modified CNT system could quantify zinc in various circumstances, being an additional possible source of information for both the diagnosis and judgement of treatment efficacy. When compared to different works in the literature (Table 3), the zincon-modified CNT showed great potential to be applied as zinc sensor. Investigating current studies, only a few were able to quantify zinc in saliva or urine, and most of them were performed in serum or other fluids, needing an invasive procedure. Our system also presented competitive results when compared to commercial colorimetric zinc rapid test for biological fluids, including urine and saliva, where the quantification is from 200 ng·mL^−1^ (Zinc Assay Kit—abcam 102507), ~2-folder lower than the zincon-modified CNT demonstrated here. Our engineered electrode demonstrated simplicity, low-cost, specificity and great ability to detect and quantify free Zinc.

## 4. Conclusions

Here, we report a new generation of organic/inorganic nanocarbon based sensing material using a striking synthetic approach capable of quantify mobile zinc ions in biofluids, i.e., saliva and urine. Currently, time consuming, and costly techniques requiring highly qualified personnel and sizable machines are available to measure and monitor metal level. Given the specific binding between zincon and zinc, and that a cheap and practical biosensing platform is preferred, carbon nanotubes were chosen and chemically modified to attach zincon onto the inert and conductive surface. The electro-capability of this sensing materials for zinc quantification was unprecedented with a LOD of ~20 and 30 ng·mL^−1^ in urine and saliva, respectively, a linear range of detection of 125–1000 ng·mL^−1^ and proven to be selective towards this bivalent cation. The zincon-modified CNT system is a specific, fast, simple, and non-expensive alternative for the quantification of zinc in easily body fluids. Given the clinical importance of measuring free zinc ions, the development of this composite nanomaterials points the way to removing a key technical bottleneck for easy, low cost and fast detection of disease biomarkers and could offer a route to cost reduction and lowering hurdles to more widespread adaptation of point of care testing.

## Figures and Tables

**Figure 1 nanomaterials-12-04431-f001:**
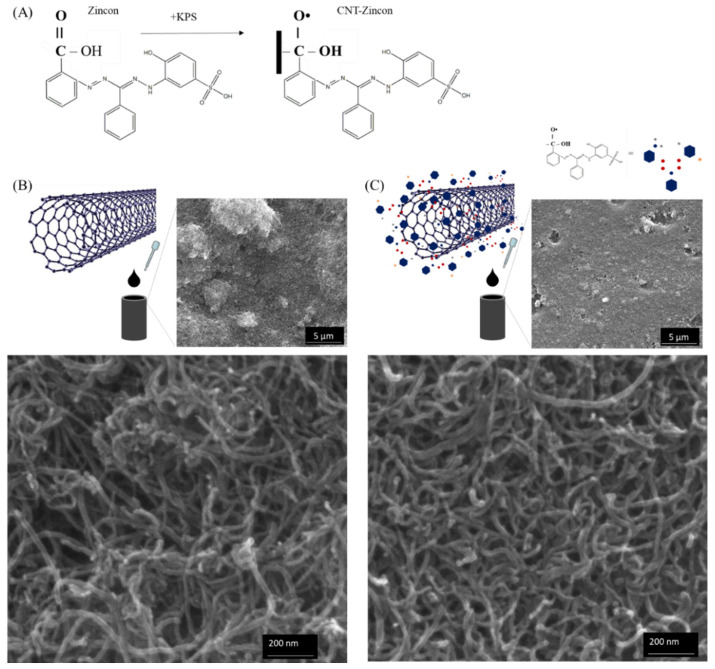
(**A**) Suggested mechanism of covalent grafting facilitated by KPS and, schematic of (**B**) CNT and (**C**) zincon-modified CNT followed by SEM images (low and high magnitude), where hexagon, gray dots, blue dots, red dots, and star represent aromatic group, oxygen element, carbon element, nitrogen element and Sulfonic acid, respectively.

**Figure 2 nanomaterials-12-04431-f002:**
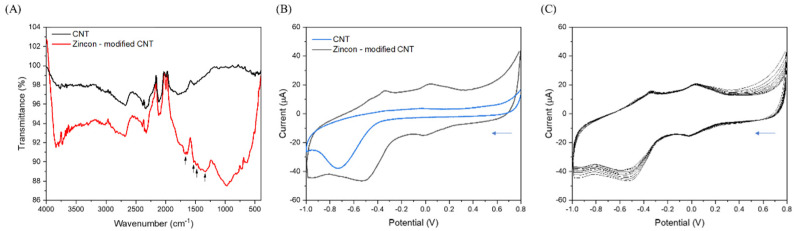
(**A**) FTIR spectrum of CNT and zincon–modified CNT, where black arrows show the zincon peaks. (**B**) CV showing the reduction and oxidation peaks of zincon present on CNT after grafting process. The zincon–modified CNT showed good stability after (**C**) 10 CVs on tris–HCl (pH 7.5), scan rate 10 mV/s. The blue arrows mean the direction of the CV from + to −.

**Figure 3 nanomaterials-12-04431-f003:**
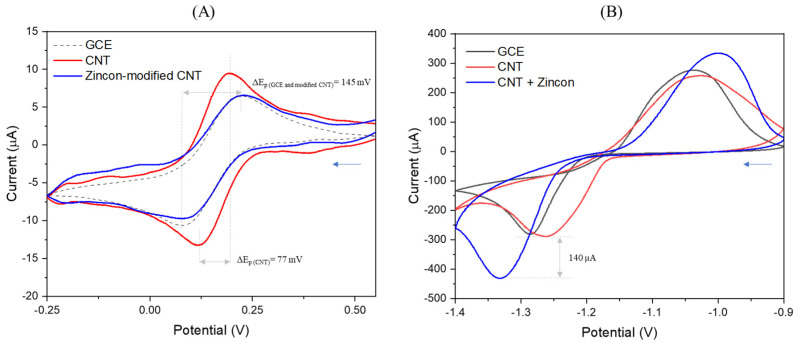
CVs for GCE, CNT and zincon–modified CNT in (**A**) 1 mM K_4_Fe(CN)_6_ in a 0.1 M KCl and (**B**) tris HCl (pH 7.5) containing 0.5 mg·mL^−1^ of zinc. The blue arrows mean the direction of the CV from + to −.

**Figure 4 nanomaterials-12-04431-f004:**
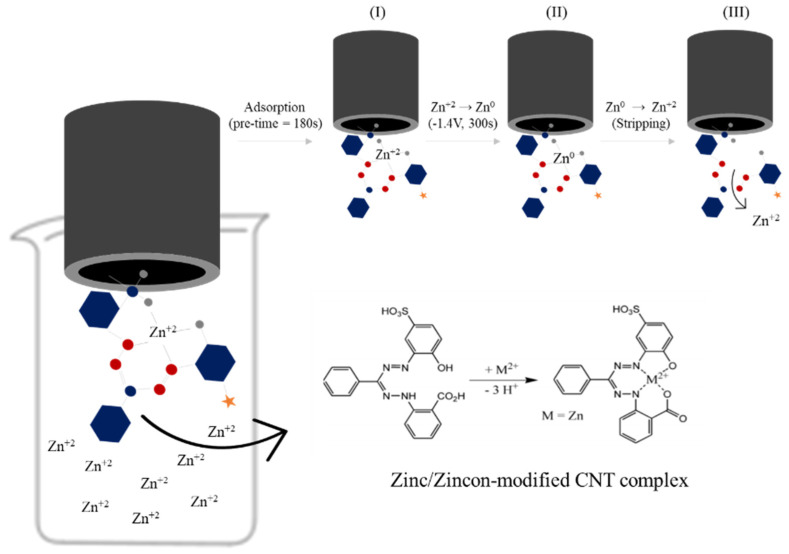
Schematic of complexation mechanism of zincon–modified CNT and Zn^+2^; (I) adsorption, (II) reduction and (III) stripping step. The hexagon, gray dots, blue dots, red dots, and star represent aromatic group, oxygen element, carbon element, nitrogen element and Sulfonic acid, respectively.

**Figure 5 nanomaterials-12-04431-f005:**
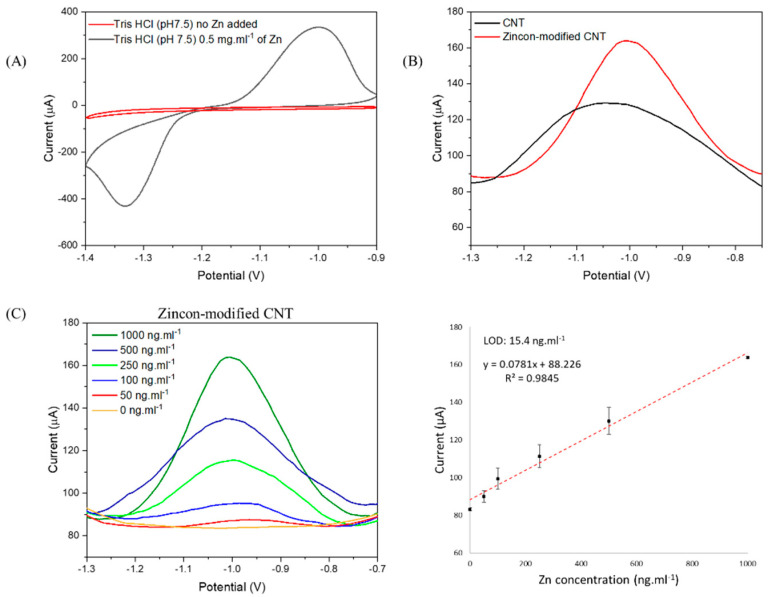
Study of the electrochemical detection of Zn^+2^ on 0.1 M Tris–HCl (pH 7.5) solution. (**A**) CV in absence and presence of 0.5 mg·mL^−1^ of Zn^+2^. (**B**) SWV showing the increase of current after grafting of zincon on CNT in Tris–HCl solution containing concentration of Zn^+2^ = 1 µg·mL^−1^, and (**C**) SWV of zincon–modified CNT varying Zn^+2^ concentration from 0 to 1000 ng·mL^−1^ and its linear fit (n = 3).

**Figure 6 nanomaterials-12-04431-f006:**
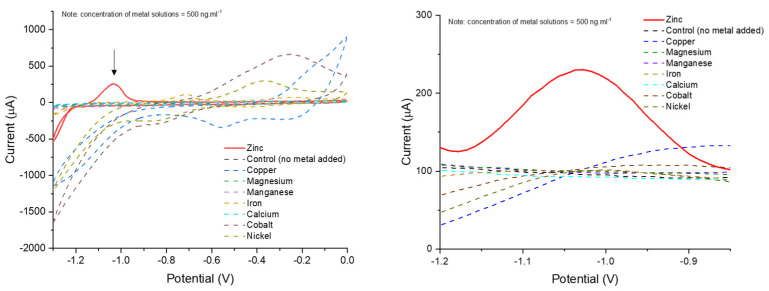
CV in absence and presence of 0.5 mg·mL^−1^ of bivalent metals in 0.1 M Tris–HCl (pH 7.5) solution. SWV showing the increase of current after grafting of zincon on CNT in Tris–HCl solution containing concentration of Zn^+2^ and other metals at a concentration of 0.5 mg·mL^−1^.

**Figure 7 nanomaterials-12-04431-f007:**
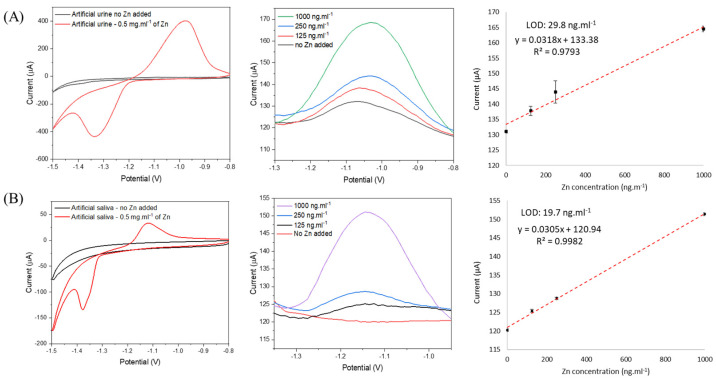
Salivary and urinary zinc quantification using zincon–modified CNT electrode. CV in absence and presence of zinc and SWV varying zinc concentration from 0 to 1000 ng·mL^−1^ in artificial (**A**) urine and (**B**) saliva, respectively (n = 3).

**Table 1 nanomaterials-12-04431-t001:** Some examples of abnormal expression of salivary and urinary zinc for different types of cancer.

Cancer	Salivary Zinc (ng·mL^−1^)		Urinary Zinc (ng·mL^−1^)		Reference(s)
	Healthy *	Unhealthy *	Healthy *	Unhealthy *	
Prostate	~270	~630	~400	~675	[7,8,9,20]
Oral	~360	~150	-	-	[10,11]
Breast	~600	~1010	~320	~602	[9,12,13,14,15]
Pancreatic	-	-	~400	~945	[9,16]
Brain	~540	~410	-	-	[17]
Lung	-	-	~570	~1500	[18]
Bladder	-	-	~470	~1000	[19]

* Mean values. Note the concentration may change depending on sex, age, and stage of the disease; - no data found in the literature.

**Table 2 nanomaterials-12-04431-t002:** Chemical composition of commercial artificial urine and saliva.

Artificial Saliva (g/L) [51]		Artificial Urine (g/L) [52]	
Sodium Chloride	1.594	Urea	25
Ammonium Nitrate	0.328	Sodium Chloride	9
Potassium Phosphate	0.636	Disodium Hydrogen orthophosphate	2.5
Potassium Chloride	0.202	Potassium dihydrogen orthophosphate	2.5
Potassium Citrate	0.308	Ammonium chloride	3
Uric Acid Sodium Salt	0.021	Creatinine	2
Urea	0.198	Sodium sulphite	3
Lactic Acid Sodium Salt	0.146		
Mucin	5		

**Table 3 nanomaterials-12-04431-t003:** Comparison of reported electrochemical quantification of zinc in different body fluids.

Strategy	Range of Detection (ng·mL^−1^)	LOD (ng·mL^−1^)	Disadvantages	Sample	Ref.
Bismuth-Graphene Oxide	20–8000	6 (buffer)	In situ bismuth co-deposition/Zinc needs extra extraction steps	Seminal Fluid	[67]
Zincon exfoliated graphite	250–1500	5 (buffer)	Zinc needs extra extraction steps	Serum	[37]
Nafion-Gold electrode	180–2500	18 (buffer)	Zinc needs extra extraction steps	Serum	[68]
Polyethyleneimine, poly (sodium 4-styrenesulfonate), and mercury nitrate on Carbon fibers	20–2000	9 (buffer)	Use of mercury compound (toxic)	Blood and Urine	[69]
Zincon-modified CNT	**125–1000**	**15 (buffer)** **20 (urine)** **30 (saliva)**		Saliva and Urine	This work

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
