# Peer review of "Zincon-Modified CNTs Electrochemical Tool for Salivary and Urinary Zinc Detection"

_nanomaterials, 2022, doi:10.3390/nano12244431_

Round 1

Reviewer 1 Report

1. In the Introduction, the authors should add more references. Like "Zinc (Zn) is one of the most abundant trace elements in the body and it is essential for 31 more than 300 enzyme functionalities"(Ref?). and " Zinc plays a crucial role to intracellular communi- 32 cation, immune system, homeostasis, apoptosis, DNA repair and replication, balancing 33 oxidative stress, and the aging process"(Ref?). Salivary and/or urinary zinc 36 level is abnormal in a variety of cancer.(Ref?).........,

2. In the material method section, the Authors should add one more subheading about "Materials and chemicals", There should be all information about materials and used chemicals which is important for the reproduction of this work. 

3. The high-magnitude SEM images do not give any more information. Authors should be denoted in the image about the interaction between zincon with CNT. Here represented Figure 1B and 1C no give any different information. 

4.  "Finally at higher magnification, the morphology of the CNTs remains similar after zincon polymerization, without any alterations of the tubular structure of CNTs (Figure 1B)" . The authors should add more discussion about the mechanism of why it is happening after coating. Authors should check the wide of CNT, there will be some changes. 

5. Figure 5 quality is very poor. Also How the authors acquired the data 5B, 5C, and its linear fit (n=2) should clearly discuss in the text.  

6. Table 2, should include references. 

7. Authors should add a selectivity graph, other positive and negative ions interaction data, etc. otherwise it is not validated for a point care tool.  

8.  Title should change, need more information about this work.  

Author Response

Reviewer 1

Thank you for your comments.

  1. In the Introduction, the authors should add more references. Like "Zinc (Zn) is one of the most abundant trace elements in the body and it is essential for 31 more than 300 enzyme functionalities"(Ref?). and " Zinc plays a crucial role to intracellular communi- 32 cation, immune system, homeostasis, apoptosis, DNA repair and replication, balancing 33 oxidative stress, and the aging process"(Ref?). Salivary and/or urinary zinc 36 level is abnormal in a variety of cancer.(Ref?).........,

RE: We added the references   

  1. In the material method section, the Authors should add one more subheading about "Materials and chemicals", There should be all information about materials and used chemicals which is important for the reproduction of this work. 

RE: Section added.

  1. The high-magnitude SEM images do not give any more information. Authors should be denoted in the image about the interaction between zincon with CNT. Here represented Figure 1B and 1C no give any different information.  Finally at higher magnification, the morphology of the CNTs remains similar after zincon polymerization, without any alterations of the tubular structure of CNTs (Figure 1B). The authors should add more discussion about the mechanism of why it is happening after coating. Authors should check the wide of CNT, there will be some changes.

RE: The SEM analysis is only a visual observation and only proved that the CNT morphology appears to be unchanged after the chemical, i.e. no obvious breakage, no tubular transformation etc. We were not expecting to observe difference between figure 1B and 1C as we choose this treatment for its advantage to enable the attachment of zincon without affecting the morphology of the carbon nanostructures. We believe that pristine CNTs are tangled together, due to the van der Waals attractions between tubes. After grafted with zincon, the materials because of the sulfonic group of the zincon, have a much better dispersibility than pristine CNTs.  We clarified this in the manuscript. We added the measurement of diameter using ImageJ software to the discussion.

  1. Figure 5 quality is very poor. Also How the authors acquired the data 5B, 5C, and its linear fit (n=2) should clearly discuss in the text.  

RE: we apologize and improved the quality of the figure. The acquisition of data for Figure 5B and 5C is explained in the methodology section, line 137 to 141. We added the information about linear fit in line 141.  

  1. Table 2, should include references. 

RE: All references were already included in now table 3. Please see picture below.

  1. Authors should add a selectivity graph, other positive and negative ions interaction data, etc. otherwise it is not validated for a point care tool.  

The interaction was evaluated with artificial urine and saliva. Both solutions contain components that could interact with the Zincon based electrode.  The artificial saliva was purchased from Pickering (catalog number 1700-0305). The artificial urine was purchased from Aldon (# IS5080). We added information about the composition in table 2 in the material section. Additionally, we runt an interference study to confirm the selectivity of this sensing surface towards zinc (Figure7)

  1. Title should change, need more information about this work.  

RE: Title was modified and can be read as “Zincon-modified CNTs electrochemical tool for salivary and urinary zinc detection”.

Reviewer 2 Report

The paper reports the fabrication of a label free electrochemical sensor based on zincon-modified carbon nanotubes for the detection and quantification of Zn ion as a point of care tool in saliva and urine samples. The manuscript describes the fabrication process of the material, its characterization and its application as electrochemical sensors, first in lab solutions and later in real samples.

The paper seems interesting; however, I am not sure that the use of only commercial carbon nanotubes as nanomaterial is completely enough to justify the publication of the paper in the Nanomaterials journal, moreover when the quality of the actual version of the manuscript is not too much good, mainly for the electrochemical part (as it can be seen below). That is why I recommend rejection of the paper.

General Remarks

Some grammar and spelling errors in the text should be corrected.

Authors affiliation is not complete.

Specific Remarks

Authors should specify well the nomenclature for the analyte: sometimes they use Zn, Zinc(II), Zn+2 (not accepted by IUPAC), and so on. They should unify all the names.

Introduction section:

The novelty of the paper should be well stressed, here and in the Abstract section. Authors should differentiate well their piece of research from others published in literature.

Experimental section should be further completed with relevant information. Some examples are:

- Diameter of the GCE electrode should be stated. All the reagents employed in the piece of research as well as their purity when possible should be included in this section.

- The ‘pre time’ experimental parameter should be better and properly addressed as ‘conditioning time’ or ‘adsorption time’, etc.

- Why do the authors test the experiments by duplicate when from the analytical point of view it is better recommended to do it three times at least?

Results and discussion section

This section should be split on different subsections cor clarity.

- The rest of peaks in Figure 2A should be indexed as well and if possible within the plot. Check the indexation of peaks in this figure as well; I think there is some error in it. Besides, there are four peaks assigned to zincon, but only three appears in the text.

- In figures 2B and 2C, an arrow should indicate the sense of the scans. It is supposed they are done from 0.3 to -1 V and viceversa, but it is not clear.

- In the text corresponding to Figure 3, it should be better to put increments of current and potential together with % of variation than full values. Potential values should be checked as well for each peak. Arrow also in figure 3 for knowing the scan sense. Figure 3 should be provided with enhanced resolution.

- The ‘pre-time’ of 180 s, is it applied at OCP or at a certain potential?

- Figure 5 is difficult to see. I do not understand why the authors obtained so high R2 for such poor fitting like the one plotted in Figure D (red line)?

- Three replicates should be done instead of 2.

- Stability, reproducibility and repeatability studies are missing.

- What about an interferent study? Nothing is said about that.

- I do not understand why they do not use standard addition method to quantify Zn2+ in saliva. According to Figure A, there is analyte in the real sample before spiking. This value would be the one they are able to 'detect' and quantify and not the first spiked value. The fitting in the calibration plot of saliva is also confusing: good R2 value for poor fitting (red line).

- References should be included in Table 2. They should justify why the lowest measurable detection value is a quality analytical parameter. I think it is better to use the sensitivity, linear range and/or LOD. In fact, authors have not tested values lower than 50 ng/mL and perhaps they could. Moreover, for ref 44, second line, the lowest measurable detection (that I think is quantification) is 32.5 ng/ml and not 250.

Conclusions section is poor. It is much more a summary than real conclusions. Regarding the first statement in this section, I think the sentence in lines 283-287 is better for this section in order to change sentence number one here that it is too much hard for the evidences given in the paper. The values reported in Conclusions section should be revised. If the specificity of the sensor is so much high, as authors states (with no evidences from a interferences study), and if they dilute 50% real samples there should not be much differences between buffer and sample study. Authors also mix detection with quantification throughout the whole paper and makes it very difficult to follow.

References: revise refs. 1 and 12.

Author Response

Reviewer 2

Thank you for your comments.

  1. Some grammar and spelling errors in the text should be corrected.

RE: The manuscript was reviewed, and the grammar corrected.

  1. Authors affiliation is not complete.

RE: We added affiliation.

  1. Authors should specify well the nomenclature for the analyte: sometimes they use Zn, Zinc(II), Zn+2 (not accepted by IUPAC), and so on. They should unify all the names.

RE: We unified all terms as zinc (Zn), and for mobile zinc ions as Zn2+

  1. The novelty of the paper should be well stressed, here and in the Abstract section. Authors should differentiate well their piece of research from others published in literature.

RE: As suggested we emphasized the original work in the abstract and introduction.

  1. Diameter of the GCE electrode should be stated. All the reagents employed in the piece of research as well as their purity when possible, should be included in this section.

RE: Purity of chemicals was added. Diameter was already mentioned in the text (5 mm), but we highlighted the information.

  1. The ‘pre time’ experimental parameter should be better and properly addressed as ‘conditioning time’ or ‘adsorption time’, etc.

RE: We clarified and added adsorption time to the manuscript.

  1. Why do the authors test the experiments by duplicate when from the analytical point of view it is better recommended to do it three times at least?

RE: We performed all electrochemical experiments in triplicates. The information was corrected in the manuscript.

  1. This section should be split on different subsections cor clarity.

RE: as suggested, the section was split into materials characterization and electrochemical behavior 

  1. The rest of peaks in Figure 2A should be indexed as well and if possible within the plot. Check the indexation of peaks in this figure as well; I think there is some error in it. Besides, there are four peaks assigned to zincon, but only three appears in the text.

RE: we indexed the other peaks related to CNTs and corrected the peaks associated with the zincon.  

  1. In figures 2B and 2C, an arrow should indicate the sense of the scans. It is supposed they are done from 0.3 to -1 V and viceversa, but it is not clear.

RE: We clarified that the scan was done from +0.8 to -1V with arrow.  

  1. In the text corresponding to Figure 3, it should be better to put increments of current and potential together with % of variation than full values. Potential values should be checked as well for each peak. Arrow also in figure 3 for knowing the scan sense. Figure 3 should be provided with enhanced resolution.

RE: Thank you, as suggested we improved the resolution added an arrow. The % of values (current and potential) were incorporated in the manuscript as well as in the figure.

  1. The ‘pre-time’ of 180 s, is it applied at OCP or at a certain potential?

RE: No potential was applied. The info was added to the text.

  1. Figure 5 is difficult to see. I do not understand why the authors obtained so high R2 for such poor fitting like the one plotted in Figure D (red line)?

RE: We apologized and reorganized the figure. About the linear regression, we repeated the fitting with the origin software, and the exact same data were obtained. We would be very happy to provide all the raw data if requested. 

  1. Three replicates should be done instead of 2.

RE: This information was wrongly added in the first submission, and we corrected since experiments were all done in triplicates.

  1. Stability, reproducibility and repeatability studies are missing.

Reproducibility and repeatability have been studied with at least 3 sensors measured 3 independent times, it was clarified in materials and methods.  The ink is an inorganic/organic hybrid material which is stable in physiological conditions and not prone to irreversible denaturation. We runt a stability study after storing the zincon-CNT ink at room temperature over a 6-month period and performed a CV before and after and observed no loss in electrochemical activity. Furthermore, another crucial feature of using zincon on carbon is not only the binding affinity but the low cost of the sensing element, which enables the sensor to be identified as a disposable point of care diagnostic tools.

  1. What about an interferent study? Nothing is said about that.

RE: The electrochemical study was evaluated in artificial urine and saliva. Both solutions contain components that could interact with the Zincon based electrode.  The artificial saliva was purchased from Pickering (catalog number 1700-0305). The artificial urine was purchased from Aldon (# IS5080). We added information about the composition in table 2 in the material section. Other components such as methylparaben (0.1%), Alizarin Yellow (0.0033%), Thymol (0.0017%)) were found in the composition. Reference: https://www.aldon-chem.com/sds/urine-artificial-set-of-4.pdf%22. Additionally, we runt an interference study to confirm the selectivity of this sensing surface towards zinc (Figure 7).

  1. I do not understand why they do not use standard addition method to quantify Zn2+ in saliva. According to Figure A, there is analyte in the real sample before spiking. This value would be the one they are able to 'detect' and quantify and not the first spiked value. The fitting in the calibration plot of saliva is also confusing: good R2 value for poor fitting (red line).

We used a standard approach to assess the performances in body fluids. In saliva or urine (Figure 6A), CV with and without any Zinc2+ enables to identify the oxidation peak at the electrode in presence of zincon. Then, we studied the SWV response currents of our Zincon/CNT system at this potential with different concentrations of Zn2+. This method is standard and demonstrates that the oxidation current of Zn adsorbed on the electrode increased with the increase of Zn2+. We fitted the data and observed that the peak currents (i, μA) increased linearly with Zn2+over the concentration range. As previously explained in previous question (13) we repeated the fitting with the origin software, and as we cannot manipulate the data during the regression, the exact same value for R was obtained, but we would gladly provide the raw data if requested. 

  1. References should be included in Table 2. They should justify why the lowest measurable detection value is a quality analytical parameter. I think it is better to use the sensitivity, linear range and/or LOD. In fact, authors have not tested values lower than 50 ng/mL and perhaps they could. Moreover, for ref 44, second line, the lowest measurable detection (that I think is quantification) is 32.5 ng/ml and not 250.

RE: The references were included since first submission (but unfortunately disappeared during manuscript formatting. We added the table again (not as picture). The reference 37 (second line) reported 250 ng/ml, and it was not in the biofluid, but in in buffer. If we consider biofluids, the reference reported a sensitivity of 2500 ng/ml. Please see picture below (supporting info at https://www-sciencedirect-com.proxy3.library.mcgill.ca/science/article/pii/S1572665718308129#s0070). As requested, we added in Table 3, the range of detection and LOD (updated table 2) and clarified the matrices in which the measurements were performed.   

  1. Conclusions section is poor. It is much more a summary than real conclusions. Regarding the first statement in this section, I think the sentence in lines 283-287 is better for this section in order to change sentence number one here that it is too much hard for the evidences given in the paper. The values reported in Conclusions section should be revised. If the specificity of the sensor is so much high, as authors states (with no evidences from a interferences study), and if they dilute 50% real samples there should not be much differences between buffer and sample study. Authors also mix detection with quantification throughout the whole paper and makes it very difficult to follow.

RE: The conclusion was completely rewritten by taking into considerations the reviewer’s comments. It is very common to dilute biofluid to reduce viscosity (for example, in reference 37, the fluid was diluted 500x, whereas in our work 2x).  We updated the terms through the manuscript.

  1. References: revise refs. 1 and 12.

Reference 1 and 12: The book page was corrected.

Reviewer 3 Report

The work, entitled "Label free electrochemical point of care tool for salivary and urinary zinc detection” aims to design and application of a non-invasive, stable, precise, and specific zincon-modified carbon nanotube (CNT) system to detect zinc in body fluids.

-        Details on the number of samples, replicates,? should be provided. One replicate is low.

-        How can be translated/implemented in other labs?

-        Contaminants analytes? Concentrations? No mention.

-        How specific are?

-        It should be tested in blinded biological samples (~n=3)

-        How can these data be translated into the clinical setting?

Author Response

Reviewer 3

Thank you for your comments.

The work, entitled "Label free electrochemical point of care tool for salivary and urinary zinc detection” aims to design and application of a non-invasive, stable, precise, and specific zincon-modified carbon nanotube (CNT) system to detect zinc in body fluids.

  • Details on the number of samples, replicates,? should be provided. One replicate is low.

All samples were performed in triplicates. The info is in the title of the figures (n=3). The info was also added to the methodology, line 141. 

  • How can be translated/implemented in other labs?

As the system is not complex, it can be easily scalable and applied to others lab using similar electrodes and conditions or simply by moving to a 2D system such as screen-printed electrodes (SPEs) and commercial handled potentiostat.

  • Contaminants analytes? Concentrations? No mention.

The artificial saliva was purchased from Pickering (catalog number 1700-0305). The artificial urine was purchased from Aldon (# IS5080). We added information about composition in table 2 in the material section. Other components such as methylparaben (0.1%), Alizarin Yellow (0.0033%), Thymol (0.0017%)) were found in the composition. Reference:https://www.aldon-chem.com/sds/urine-artificial-set-of-4.pdf%22. Additionally, we runt an interference study to confirm the selectivity of this sensing surface towards zinc (Figure 7)

  • How specific are? It should be tested in blinded biological samples (~n=3). How can these data be translated into the clinical setting

Considering the contaminants cited above, we believe that this proof of principle study on zincon-modified CNT system exhibits a great specificity towards zinc in both artificial saliva and urine. We agree that before being implemented, other parameters and conditions must be studied. This full study is out of the scope of this paper but could in the future include not only sensor evaluation in blinded human samples such as saliva and urine fluids, translation to a 2D system  but also a comparison with conventional approaches and other electrochemical sensors.

Round 2

Reviewer 1 Report

1. In the CV graph, Figure 7A/B- only Zn, and different concentrations of Zn with urine and salivary should represent. 

  1. The high-magnitude SEM images do not give any more information. Authors should be denoted in the image about the interaction between zincon with CNT. Here represented Figure 1B and 1C no give any different information.  Finally at higher magnification, the morphology of the CNTs remains similar after zincon polymerization, without any alterations of the tubular structure of CNTs (Figure 1B). The authors should add more discussion about the mechanism of why it is happening after coating. Authors should check the wide of CNT, there will be some changes.

RE: The SEM analysis is only a visual observation and only proved that the CNT morphology appears to be unchanged after the chemical, i.e. no obvious breakage, no tubular transformation etc. We were not expecting to observe difference between figure 1B and 1C as we choose this treatment for its advantage to enable the attachment of zincon without affecting the morphology of the carbon nanostructures. We believe that pristine CNTs are tangled together, due to the van der Waals attractions between tubes. After grafted with zincon, the materials because of the sulfonic group of the zincon, have a much better dispersibility than pristine CNTs.  We clarified this in the manuscript. We added the measurement of diameter using ImageJ software to the discussion.

Q. Authors should add SEM images of Zincon only. 

  1. Authors should add a selectivity graph, other positive and negative ions interaction data, etc. otherwise it is not validated for a point care tool.  

 Q.The above question does not solve properly. In the interference study,(Figure-6) authors should include a few negative metal ions also which are often present in urine and saliva. 

Author Response

Reviewer 1 

  • In the CV graph, Figure 7A/B- only Zn, and different concentrations of Zn with urine and salivary should represent. 

The data were obtained in the presence and absence of zinc. The variation in concentration is represented in the SWV experiment. CV was applied to confirm that the peaks on SWV were attributed only to the presence of zinc and not to other elements. As noted in the CVs, zinc exhibits clearly well-defined peaks. The variation of the current in function of concentration it is not usually represented by CV when measurements are taken at the nanoscale.

  • Authors should add SEM images of Zincon only. 

Zincon is an organic salt (see fig. in word) that will be dissolved to chemically react with carbon.  Taking a SEM picture of a salt, will not improve or add any value to the understanding of the structure of the electrode surface.

  • Authors should add a selectivity graph, other positive and negative ions interaction data, etc. otherwise it is not validated for a point care tool. Q: The above question does not solve properly. In the interference study,(Figure-6) authors should include a few negative metal ions also which are often present in urine and saliva.

Zincon is a well-known and non-selective reagent for the photometric determination of cations. It will form a complex or chelate positively charged metal ions (see figure below) and was used particularly for the determination of Zn2+, Cu2+, and Co2+. Given the mechanism upon which zincon reacts with metal ions, only bimetallic cations were chosen. Ca, Cu, Co, Mg, Mn, and Fe are the most common metals in the body and also in body fluids. We clarified this and added the references to make it clear to the reader.

Reviewer 2 Report

The paper has been enhanced further. However there are still some questions that should be solved out.

1) Authors say that they do not apply a potential value during 'pre-time' (adsorption time). They should specify whether the electrical connections in the electrochemical cells are stablished or not. If they are, they are under OCP; if not, they are not applying any potential value. This fact should specify in the Materials and Methods section.

2) They do not give an answer about the signal appearing in Figure 7 A  (SWV) regarding the 'no Zn added' voltammogram. It seems that there is some amount of Zn2+ in the artificial urine.

3) No clear information/results about stability, reproducibility and repeatability are included in the paper and it should. If authors have performed these studies, that information should be provided.

4) Regarding the fitting values provided in Figure 5D and Figure 7 A-B, I still do not agree with the R2 values. It is a matter of fact that if 2 of the 4 points of the calibration curve, considering also the errors values in Figure 7A are up the regression line (red line), for having the fitting of 0.99 at least there should be some point below the fitting line or with errors below that fitting line (errors for points 1 and 4 seems to be rather low). For example, taking approximately the values from the calibration curve reported in Figure 7A (0, 131; 135, 137.5; 250, 144; and 1000, 164.5 in ng/mL, uA for points 1 to 4, respectively) and performing a simple linear regression in Excel, no forcing the fitting line to pass by the intercept, the fitting is y=0.0321x+113.15 (R2 = 0.9791). As authors can see, the values of the fitting equation are very close to theirs (I mean slope and intecept; errors can come for not having the exact values), and the fitting is the expected one and not as good as reported. If we force the fitting line to pass by the intercept, we obtain a poorer fitting with R2 = 0.9629. I suggest do not forcing the intercept and to perform the fitting with other statistics software to check.

Author Response

Montreal, November 28 2022

Dear Dr. Liu,

Please find our manuscript “Zincon-modified CNTs electrochemical tool for salivary and urinary zinc detection” for consideration in Nanomaterials.  This manuscript has been previously considered as manuscript number nanomaterials-1996276. We have addressed the reviewers’ comments and provide detailed point-by-point responses below.

We certify that this manuscript, or its contents in some other form, has not been published previously by any of the authors and/or is not under consideration for publication in another journal at the time of submission. All the authors have seen and approved the submission of the manuscript.

I am looking forward to hearing from you shortly.

Yours sincerely,

Geraldine Merle

Reviewer 2

The paper has been enhanced further. However there are still some questions that should be solved out.

  • Authors say that they do not apply a potential value during 'pre-time' (adsorption time). They should specify whether the electrical connections in the electrochemical cells are stablished or not. If they are, they are under OCP; if not, they are not applying any potential value. This fact should specify in the Materials and Methods section.

The electrical connections were not connected during the adsorption, we clarified this in the text

  • They do not give an answer about the signal appearing in Figure 7 A  (SWV) regarding the 'no Zn added' voltammogram. It seems that there is some amount of Zn2+ in the artificial urine.

We have asked the manufacturer about the composition and in addition to the component in the table, they mentioned to add thymol and Tegosept as preservative. Thymol would appear around 0.5V vs SCE on carbon electrode whereas Tegosept at 0.7-0.8V vs SCE.  The analysis without Zinc2+ in urine was repeated and similar result was obtained. One must keep in mind that each manufacturer has their own “recipe” so although we agree that there is certainly a contaminant present in this artificial urine (maybe a purity issue with materials they used), it is rather difficult to pinpoint the contaminant without having a clear answer from the manufacturer about the materials and exact composition. Nevertheless, it is important to note the increase of current with incremental increase of Zinc2+ concentration in the artificial media.    

  • No clear information/results about stability, reproducibility and repeatability are included in the paper and it should. If authors have performed these studies, that information should be provided.

Repeatability is the precision determined from multiple tests done under repeatability conditions: the test is conducted by the same operator, using the same equipment and laboratory within a short period of time so that neither the equipment nor the environment is likely to change significantly. We have tested repeatability and we can define it by the repeatability standard deviation, which can be measured by the square root of the variance. For the buffer, saliva and urine, the repeatability standard deviation was 7.07 µA, 0.55 µA and 3.28 µA, respectively.

Reproducibility is the precision determined from multiple tests done under reproducibility conditions: the test is conducted in different laboratories with different operators and different environmental conditions. While two students worked on the synthesis and electrochemical characterization at different times, we cannot mention that the reproducibility was studied.

Stability describes the variation in detection signals during a period of long-term storage and can be defined by see word document for equation.Electrodes were kept and measured after 1 year, and based on the above equation, the stability was ~65%. We added repetability and stability values in the materials and methods, and discussion.

  •  Regarding the fitting values provided in Figure 5D and Figure 7 A-B, I still do not agree with the R2 values. It is a matter of fact that if 2 of the 4 points of the calibration curve, considering also the errors values in Figure 7A are up the regression line (red line), for having the fitting of 0.99 at least there should be some point below the fitting line or with errors below that fitting line (errors for points 1 and 4 seems to be rather low). For example, taking approximately the values from the calibration curve reported in Figure 7A (0, 131; 135, 137.5; 250, 144; and 1000, 164.5 in ng/mL, uA for points 1 to 4, respectively) and performing a simple linear regression in Excel, no forcing the fitting line to pass by the intercept, the fitting is y=0.0321x+113.15 (R2 = 0.9791). As authors can see, the values of the fitting equation are very close to theirs (I mean slope and intecept; errors can come for not having the exact values), and the fitting is the expected one and not as good as reported. If we force the fitting line to pass by the intercept, we obtain a poorer fitting with R2 = 0.9629. I suggest do not forcing the intercept and to perform the fitting with other statistics software to check.

We followed the reviewer instructions and used excel software. We obtained R2 value of 0.9845. We reviewed and updated the graphs in Figures 5 and 7.

Reviewer 3 Report

The authors have performed suggested corrections.

Author Response

Thank you for your comments. 

Round 3

Reviewer 1 Report

  1. "Authors should add a selectivity graph, other positive and negative ions interaction data, etc. otherwise it is not validated for a point care tool."  Till this problem not fixed.

Reviewer 2 Report

After revising the new version of the manuscript, I consider it is suitable for publication in its present form. However, please, check the form they express the repeatibility. I think it is better to provide the average value plus/minus the standar deviation or as relative standard deviation in percentage.

Author Response

Thank you for you comments. We now added the standard deviation to the repeatability.